# Non-Toxicological Role of Aryl Hydrocarbon Receptor in Obesity-Associated Multiple Myeloma Cell Growth and Survival

**DOI:** 10.3390/cancers15215255

**Published:** 2023-11-01

**Authors:** Jonathan D. Diedrich, Craig E. Cole, Matthew J. Pianko, Justin A. Colacino, Jamie J. Bernard

**Affiliations:** 1Department of Pharmacology and Toxicology, Michigan State University, East Lansing, MI 48824, USA; diedri14@msu.edu; 2Department of Medicine, Division of Hematology/Oncology, Michigan State University, East Lansing, MI 48910, USA; colecrai@msu.edu; 3Karmanos Cancer Institute, McLaren Greater Lansing, Lansing, MI 48910, USA; 4Department of Medicine, Michigan State University, East Lansing, MI 48824, USA; 5Department of Internal Medicine, Division of Hematology/Oncology, University of Michigan, Ann Arbor, MI 48109, USA; mpianko@med.umich.edu; 6Department of Nutritional Sciences, University of Michigan, Ann Arbor, MI 48109, USA; colacino@umich.edu; 7Department of Environmental Health Sciences, University of Michigan, Ann Arbor, MI 48109, USA

**Keywords:** aryl hydrocarbon receptor, obesity, multiple myeloma, bone marrow adipocytes, tumor microenvironment

## Abstract

**Simple Summary:**

Multiple myeloma (MM) patients with obesity progress faster from myeloma precursor disorders compared to lean individuals. Understanding the biological contribution of increased adipocyte numbers in the marrow space which supports MM cell growth and survival is important for identifying novel therapeutic options to slow MM progression. We confirmed in a panel of six human MM cell lines (MM.1S, MM.1R, KMS11, H929, RPMI8226, and U266B1) that AhR activity is suppressed by exposure to secreted factors from bone marrow adipocytes. A knockdown of AhR in MM cells increased cellular growth. A computational analysis of the survival data showed that high AhR expression and low AhR target gene expression are associated with worse outcomes in MM, demonstrating that the inhibition of AhR activity supports MM cell growth and survival.

**Abstract:**

Obesity is not only a risk factor for multiple myeloma (MM) incidence, but it is also associated with an increased risk of progression from myeloma precursors—monoclonal gammopathy of undetermined significance—and smoldering myeloma. Adipocytes in the bone marrow (BMAs) microenvironment have been shown to facilitate MM cell growth via secreted factors, but the nature of these secreted factors and their mechanism of action have not been fully elucidated. The elevated expression of aryl hydrocarbon receptor (AhR) is associated with a variety of different cancers, including MM; however, the role of AhR activity in obesity-associated MM cell growth and survival has not been explored. Indeed, this is of particular interest as it has been recently shown that bone marrow adipocytes are a source of endogenous AhR ligands. Using multiple in vitro models of tumor–adipocyte crosstalk to mimic the bone microenvironment, we identified a novel, non-toxicological role of the adipocyte-secreted factors in the suppression of AhR activity in MM cells. A panel of six MM cell lines were cultured in the presence of bone marrow adipocytes in (1) a direct co-culture, (2) a transwell co-culture, or (3) an adipocyte-conditioned media to interrogate the effects of the secreted factors on MM cell AhR activity. Nuclear localization and the transcriptional activity of the AhR, as measured by *CYP1A1* and *CYP1B1* gene induction, were suppressed by exposure to BMA-derived factors. Additionally, decreased AhR target gene expression was associated with worse clinical outcomes. The knockdown of AhR resulted in reduced *CYP1B1* expression and increased cellular growth. This tumor-suppressing role of *CYP1A1* and *CYP1B1* was supported by patient data which demonstrated an association between reduced target gene expression and worse overall survival. These data demonstrated a novel mechanism by which bone marrow adipocytes promote MM progression.

## 1. Introduction

Multiple myeloma (MM), an incurable malignancy of bone marrow-residing plasma cells, has been associated with age, race, and obesity [1,2,3,4,5]. MM is currently the second most common hematological malignancy in high-income countries. The etiology of this disease is complex and incompletely understood. Obesity is a risk factor for MM incidence, and MM patients with obesity have increased risks for the progression of MM pre-cursors (the monoclonal gammopathy of undetermined significance (MGUS)) and smoldering myeloma (SMM) to active MM [6,7,8,9,10,11]. In addition, MM patients with obesity have poorer outcomes, including increased resistance to chemotherapies and worse overall survival rates [12,13,14]. Therefore, understanding the mechanisms by which obesity contributes to disease pathogenesis will lead to novel targets for intervention.

Advanced age and obesity are characterized by the accumulation of adipocytes within bone marrow cavities [15,16,17], where plasma cells reside. Bone marrow adipocytes (BMAs) have the propensity to secrete a multitude of factors, such as lipids, growth factors, hormones, and peptides, to induce signaling networks, alter neighboring cellular metabolisms, and stimulate cell growth across a variety of bone-tropic cancers [18,19,20,21,22,23,24]. The accumulation of BMAs has been associated with decreased bone mass via reduced osteogenesis [25,26,27] and increased chronic inflammation and excess cytokine production within the marrow space [20,28,29,30], which provides a suitable microenvironment for tumor initiation and growth. Exposure to various pro-inflammatory cytokines associated with increased adiposity, such as interleukin-6 (IL-6), stimulates differentiation, proliferation, and apoptosis in MM cells. Many of these cytokines are also elevated in the sera of MM patients [31,32,33,34,35]. Additionally, MM cells stimulate lipolysis and the release of lipids from adipocytes which can be taken up by myeloma cells as fuel to proliferate and survive in bone marrow niches [36]. The molecular mechanisms behind obesity and increased risk have been investigated by few studies. Li et al. demonstrated in a pre-clinical model that adipocyte-secreted angiotensin II induces acetyl co-A synthase 2 (ACCS2), which interacts with and stabilizes the oncogene, interferon regulatory factor 4, to promote MM growth [37]. However, pre-clinical MM models (1) are technically challenging, and (2) they do not always recapitulate human disease, which is why knowledge gaps in the mechanisms of progression of pre-cancers to active myeloma remain.

Aryl hydrocarbon receptor (AhR) is a ligand-mediated transcription factor critical for xenobiotic metabolism via the transcriptional regulation of Phase I and Phase II metabolizing enzymes [38,39,40], such as the cytochrome P450s, CYP1A1 and CYP1B1, and the glutathione-S-transferases. Upon the binding of exogenous or endogenous ligands, AhR will dissociate from its cytoplasmic complex and translocate to the nucleus, dimerize with its nuclear cofactor, aryl hydrocarbon receptor nuclear translocator (ARNT), and recognize xenobiotic response elements (XREs) across the genome. The activation of AhR has ligand- and cell-type-specific effects in normal physiological processes, such as development and hematopoiesis, as well as the metabolism and clearance of toxicants following exposure [39,41,42,43,44,45,46,47,48,49]. AhR has been implicated in a variety of different hematological malignancies, including MM [39,50,51,52,53,54]. Occupational exposure to dioxins, including the AhR ligand 2,3,7,8-Tetrachlorodibenzo-p-dioxin (TCDD), has been linked to breast, endometrial, liver, testicular, lung, and blood cancers [55,56,57]. Conversely, diminished AhR signaling in acute and chronic myeloid leukemia (AML/CML) has been shown to drive leukemic stem cell maintenance and disease progression [58,59]. The protective role of AhR in tumorigenesis has been shown through the transcriptional regulation of tumor suppressor genes that reduce tumor burdens in mouse models of multiple tumor types [60,61,62], suggesting a paradoxical role of AhR in carcinogenesis.

Our laboratory recently demonstrated that bone marrow adipocytes metabolize tryptophan into the endogenous AhR ligands kynurenine and kynurenic acid [63]. Since AhR ligands, such as polyaromatic hydrocarbons, are potent risk factors for MM and adipocytes are an endogenous source of AhR ligands [64,65], we sought to determine if bone marrow adipocytes promote MM progression through AhR-dependent mechanisms. Using a combination of in situ and in vitro analyses, we aimed to characterize AhR expression and activity across several cells lines and investigate interactions between BMAs and MM cells with respect to adipocyte-mediated AhR activity.

## 2. Materials and Methods

### 2.1. Cell Culture and Reagents

The human multiple myeloma cell lines MM.1S, MM.1R, U266B1, and RPMI8226 were purchased from the American Type Culture Collection (ATCC; Manassas, VA, USA). The H929 and KMS11 cells were generously provided by Dr. Moshe Talpaz and Dr. Malathi Kandarpa of the University of Michigan Multiple Myeloma Tissue Repository and Biobank (Ann Arbor, MI, USA). All cells were grown in RPMI-1640 medium (ThermoFisher, Pittsburgh, PA, USA) supplemented with 10% fetal bovine serum (FBS; ThermoFisher, Pittsburgh, PA, USA) and 1% penicillin-streptomycin (Pen/Strep; Sigma, St. Louis, MO, USA). The cells were maintained in a 37 °C humidified incubator supplied with 5% CO_2_.

The primary mouse bone marrow stromal cells (mBMSCs) were kindly provided by Dr. Izabela Podgorski (Wayne State University, Detroit, MI, USA). The mBMSCs were differentiated to mature adipocytes using established protocols [21,66,67]. In brief, the mBMSCs were embedded in a 3D collagen I matrix (1:1:8 of 10 × PBS:0.1N NaOH:Collagen Type 1) using PureCol^®^ collagen type I (Advanced Biomatrix, San Diego, CA, USA) and grown to confluency. Upon confluency, cells were differentiated using an adipogenic cocktail consisting of 30% StemXVivo Adipogenic Supplement (R&D Systems, Minneapolis, MN, USA), 1 µM insulin (Sigma, St. Louis, MO, USA), and 2 µM Rosiglitazone (Cayman Chemical, Ann Arbor, MI, USA) in complete DMEM medium for 8–10 days. The mBMSCs and differentiated adipocytes were cultured in low-glucose Dulbecco’s modified eagle’s medium (DMEM; Sigma, St. Louis, MO, USA). The differentiated bone marrow adipocytes were washed 3 times with PBS to remove all differentiation cocktail before being used in the experiments. The mBMSCs and adipocytes were maintained in a 37 °C humidified incubator supplied with 5% CO_2_.

### 2.2. In Vitro Models

Direct Co-culture (Adipo DCC): Mouse bone marrow stromal cells were embedded in collagen I, plated in 6-well plates, and differentiated into bone marrow adipocytes. We plated 2 × 10^6^ MM cells in 3 mL of a 1:1 ratio of complete DMEM and RPMI-1640 in co-culture with adipocytes (Adipo DCC) or on top of the collagen I matrix without adipocytes as the control (Alone). Cells were cultured for 48 hours unless otherwise specified. Myeloma cells in the suspension were removed and centrifuged at 300× *g* for 5 min. Cells were then washed 3 times with cold PBS and resuspended in RLT lysis buffer (Qiagen, Hilden, Germany) to isolate RNA or in RIPA lysis buffer (ThermoFisher, Pittsburgh, PA, USA) for the protein analysis.

Transwell Co-culture (Adipo TW): Mouse bone marrow stromal cells were embedded in collagen I, plated in 6-well plates, and differentiated into bone marrow adipocytes. Myeloma cells were then seeded on the top layer of a 0.4 µm pore-sized Transwell insert (Corning Costar, Corning, NY, USA). The porous membrane allowed crosstalk between the multiple myeloma cells and the differentiated adipocytes to facilitate the transfer of lipids and proteins but prevent direct cell-to-cell contact. 2 × 10^6^ MM cells were seeded either in the transwell insert in the co-culture with the adipocytes (Adipo TW) or in the ‘alone’ condition (Alone) in 3 mL of a 1:1 ratio of complete DMEM to complete RPMI-1640. After 48 h, the cells were removed from the top layer of the transwell insert, centrifuged at 300× *g* for 5 min, washed 3 times with PBS, and resuspended in RLT lysis buffer for the downstream RNA analyses or in RIPA lysis buffer for the protein detection.

Adipocyte-Conditioned Media (Adipo CM): mBMSCs were differentiated into adipocytes and washed 3 times with PBS. 2 mL of serum-free DMEM was added to each well of the differentiated adipocytes and incubated for 48 hours, and then collected, aliquoted, and stored them at −80 °C. 2 × 10^6^ MM.1S cells were plated in 3 mL of a 1:1 complete RPMI-1640:serum-free DMEM (Alone) or in the adipocyte-conditioned media (Adipo CM). MM cells were grown in the presence or absence of Adipo CM for 24 hours prior to harvesting.

### 2.3. RNA Isolation and TaqMan RT-PCR Analyses

Upon collection, the MM cells were washed 3 times with PBS and lysed in RLT lysis buffer (Qiagen, Hilden, Germany). The RNA was then isolated using a RNeasy Mini Kit (Qiagen, Hilden, Germany) following the manufacturer’s protocol. cDNA was generated from 1–2 ug of the total isolated RNA using a High-Capacity cDNA Reverse Transcription Kit (ThermoFisher, Pittsburgh, PA, USA). The human AhR (Hs00169233), *CYP1A1* (Hs01054797), and *CYP1B1* (Hs00164383) gene levels were assessed using TaqMan qPCR in biological triplicates in a 96-well microplate using QuantStudio 6 Flex Real-Time PCR (Applied Biosystems). Human *TBP* (Hs00427620) was used as an internal loading control. DataAssist™ software (Version 3.0) was used for all the differential expression analyses. Bar graphs were generated using GraphPad Prism software (Version 9.4).

### 2.4. Immunoblot Analyses

The human MM cell lysates were generated using RIPA lysis buffer (ThermoScientific, Pittsburgh, PA, USA) supplemented with 1% protease and phosphatase inhibitors (ThermoScientific, Pittsburgh, PA, USA) and 1% phenylmethylsulfonyl fluoride (PMSF; Sigma, St. Louis, MO, USA). The total protein concentrations were measured using a Pierce™ BCA Protein Assay (ThermoFisher, Pittsburgh, PA, USA). Equal amounts of protein were loaded to each well of a Mini-PROTEAN TGX 4–20% SDS-PAGE gradient gel (Bio-Rad Laboratories; Hercules, CA, USA) and transferred to a Trans-Blot^®^Turbo™ Mini-size nitrocellulose membrane (Bio-Rad Laboratories; Hercules, CA, USA) using the standard protocol for the Trans-Blot Turbo Transfer System. The membranes were blocked in 5% milk and immunoblotted for AhR (1:1000), Cas9 (1:1000), GAPDH (1:1000), Histone H3 (1:1000), or Actin (1:1000). The rabbit anti-human AhR (83200S), Histone H3 (4499S), GAPDH (2118S), and mouse-anti Cas9 (14697T) were purchased from Cell Signaling Technologies (Beverly, MA, USA). The rabbit-anti human Actin was purchased from Sigma (St. Louis, MO, USA). The donkey anti-rabbit (LI-COR; 926-32213), goat anti-rat (LI-COR; 926-68076), and donkey anti-mouse (LI-COR; 926-32212) secondary antibodies were used in ratios of 1:10,000 in 1% BSA, and the fluorescent signals were measured using an LI-COR Odyssey CLx.

### 2.5. Cellular Fractionation

The MM.1S, KMS11, H929, and RPMI8226 cells were cultured alone or in direct co-cultures with differentiated adipocytes for 48 h, and the whole-cell lysates were collected. Additionally, the MM.1S and KMS11 cells were treated with 100 nM 6-Formylindolo[3,2-b]carbazole (FICZ; Sigma, St. Louis, MO, USA), 10 µM CH223191 (Sigma, St. Louis, MO, USA), or 200 µM L-Kynurenine (L-Kyn; Sigma, St. Louis, MO, USA) for 24 h. Upon harvesting, the cell lysates were fractioned using a Cell Fractionation Kit (Cell Signaling Technologies, Danvers, MA, USA) according to the manufacturer’s protocol. Equal amounts of lysate from the cytoplasmic and nuclear fractions were loaded, and the AhR localization was assessed by immunoblotting for the AhR. GAPDH was used as a cytoplasmic loading control, and Histone H3 was used as a nuclear control.

### 2.6. CRISPR Knockdown of AhR in the MM.1S Cells

The MM.1S-Cas9 empty vector and MM.1S-Cas9-AhR-sgRNA were generated as previously described [63]. Briefly, the MM.1S cells were spinfected (2000 RPMs for 2 h at 37 °C) with lentiviral particles encapsulating plasmids to generate Cas9 expression and either an empty vector (EV) or sgRNA targeting AhR (AhR KD). The MM.1S-EV and MM.1S-AhR sgRNA-expressing cells were selected using 0.5 µg. Western blot and a TaqMan qPCR were used to assess the AhR knockdown in the MM.1S cells.

### 2.7. Proliferation Assays

We plated 100,000 MM.1S-Cas9 empty vector (EV) or AhR knockdown (AhR KD) cells in a 12-well plate in 1 mL of complete RPMI-1640. The cells were scraped and counted using a Bio-Rad TC10 Automated Cell Counter (Bio-Rad, Hercules, CA, USA). The number of viable cells was recorded, and bar graphs showing the growth rates of the MM.1S-Cas9 EV and AhR KD over 72 h were plotted in GraphPad Prism from the biological triplicates.

We plated 15,000 MM.1S cells in each well of a white-bottomed 96-well plate in 50 µL of complete RPMI-1640 and 50 µL of either serum-free DMEM or adipocyte-conditioned media. After 72 h, the CellTiter-Fluor (Promega, Madison, WI, USA) reagent was added, and the fluorescence was measured according to the manufacturer’s protocol. The experiments were performed using at least technical quadruplicates, and they were performed in biological triplicates.

For the viability assays in response to the CH223191 and Clofazimine (CLF), 15,000 U266B1, MM.1S, or KMS11 cells were plated in each well of a white-bottomed 96-well plate in a total of 100 µL of complete RPMI-1640 and varying doses of CH223191 (50, 20, 10, 1, and 0.5 µM) or CLF (40, 20, 10, 1, and 0.1 µM). After 72 h, CellTiter-Glo (Promega, Madison, WI, USA) was added and the luminescence was measured according to the manufacturer’s protocol. At least technical triplicates were used across two biological replicates.

### 2.8. Agonist and Antagonist Studies

The MM.1S, MM.1R, H929, KMS11, RPMI8826, or U266B1 cells were plated in 3 mL of complete RPMI-1640 medium and allowed to settle overnight. To assess the AhR activation, 100 nM of 6-Formylindolo[3,2-b]carbazole (FICZ; Sigma, St. Louis, MO, USA) or 200 µM of L-Kynurenine (L-Kyn; Sigma, St. Louis, MO, USA) were added to the cells, and they were allowed to incubate for 24 h. DMSO and 0.5 N HCl were used as the vehicle controls for the FICZ and L-Kyn, respectively. To measure the effects of the AhR antagonism in the MM cells, 10 µM CH223191 (Sigma, St. Louis, MO, USA), 10 µM alpha-Naphthoflavone (Sigma, St. Louis, MO, USA), or 1 µM Kyn-101 (Aobious, Gloucester, MA, USA) were added to the cells, and they were incubated for 24 h. The cells were harvested and the AhR target gene expression and AhR protein levels were measured using the RNA and protein extracts, respectively.

### 2.9. Survival Plots

The microarray gene expression data from patients with multiple myeloma (GSE9782) were interrogated using the Kaplan–Meier Plotter Database [68] and selecting the gene probes for AhR (202820_at), *CYP1A1* (205749_at), and *CYP1B1* (202437_s_at). The best cutoff was selected to be automatically determined on the KM plotter interface, and the KM survival plots were downloaded for each gene based off high and low expression levels.

## 3. Results

### 3.1. MM Cell Lines Are Heterogeneous in Their AhR Expression and Response to AhR Modulators

A panel of six human MM cell lines (H929, MM.1S, MM.1R, KMS11, RPMI8226, and U266B1) were selected for the in vitro analysis of AhR expression and activity. The MM.1S, MM.1R, and KMS11 cells expressed the highest levels of AhR, whereas the H929 and RPMI8226 cells expressed lower levels of AhR (Figure 1A). AhR was absent in the U266B1 cells (Figure 1A), as others have previously shown [51]. This panel of cells was screened for AhR activity in response to the AhR agonists 6-Formylindolo(3,2-b) carbazole (FICZ; 100 nM) and L-Kynurenine (L-Kyn; 200 µM) or to the antagonists CH223191 (10 µM), alpha-Naphthoflavone (α-NF; 10 µM), and Kyn-101 (1 µM) to characterize the AhR activity across the cell lines. Independent of basal AhR protein levels, the AhR agonist treatments decreased the total AhR protein levels in all the AhR-expressing cell lines (Figure 1B), increased the nuclear protein levels in the two selected cell lines (Figure 1C, MM.1S and KMS11), and significantly induced the AhR target genes *CYP1A1* and *CYP1B1* at 24 h (Figure 1D). This was consistent with what has been observed for AhR regulation in other tissues in other cell models. Conversely, the treatments with AhR antagonists increased the total AhR protein levels (Figure 1B), decreased the nuclear AhR protein levels in the two selected cell lines (Figure 1C, MM.1S and KMS11), and suppressed *CYP1A1* and *CYP1B1* expression at 24 h (Figure 1D). The RPMI8226 cells had the most drastic increases in *CYP1A1* expression, but *CYP1B1* expression was not detected. These findings showed that the MM cell lines which expressed AhR responded to both AhR agonists and antagonists, and AhR activity was regulated by nuclear protein localization. Importantly, the AhR-null cell line U266B1 did not respond to AhR agonism or antagonism.

### 3.2. AhR Activity and Multiple Myeloma Survival

To assess the role of AhR activity in myeloma progression, publicly available gene expression data were assessed using the Kaplan–Meier Plotter database (KM plotter) [68]. Elevated *AhR* expression (Figure 2A) and the reduced expression of *CYP1A1* (Figure 2B) or *CYP1B1* (CYP11; Figure 2C) were all independently associated with worse overall prognoses (GSE9782). This suggested that the inhibition of classical AhR activity is predictive of worse clinical outcomes in MM patients.

### 3.3. Bone Marrow Adipocytes Modulated AhR Expression and Activity in the MM Cells

To determine if bone marrow adipocytes (BMAs) can modulate AhR expression and activity in MM cells, multiple in vitro models of MM cell–adipocyte interactions were performed. Mouse bone marrow stromal cells (mBMSCs) were differentiated ex vivo to mature bone marrow adipocytes, and MM cells were grown in either direct co-cultures with BMAs (Adipo DCC) or in the top layer of a transwell insert in the co-cultures with BMAs (Adipo TW). The direct co-cultures allowed cell-to-cell contact and the transfer of secreted factors between the MM cells and BMAs, mimicking the in vivo microenvironment, while the transwell model acted as a physical barrier between the MM cells and the BMAs, maintaining the crosstalk between the secreted factors but removing the cell-to-cell contact (Figure 3A). The cells with the highest basal levels of AhR (MM.1S, MM.1R, and KMS11) displayed elevated protein levels of AhR when in direct co-cultures (DCC) with adipocytes (Figure 3B). This effect was not seen in the cell lines that expressed AhR at low levels (H929 and RPMI8226), and AhR remained undetectable in the U266B1 cells (Figure 3B). At the mRNA level, the co-cultures with adipocytes resulted in only minor increases in AhR expression levels in the MM.1S and MM.1R cells, and no changes or minor decreases were observed in the KMS11, RPMI8226, and H929 cells (Appendix A), suggesting that the increased total protein (Figure 3B) was not a result of increased AhR transcription.

Characterization of the MM cell responses to agonists and antagonists demonstrated that increased total AhR protein levels were associated with reduced AhR nuclear localization and activity (Figure 1). Therefore, AhR nuclear localization and *CYP* transcriptional activity were assessed in the MM cells in direct co-cultures with adipocytes. AhR was sequestered in the cytoplasm, and reduced levels of nuclear AhR were observed in both the MM.1S and KMS11 cells grown in direct co-cultures with adipocytes at 48 h (Figure 3C). Additionally, in the low-AhR-expressing H929 and RPMI8226 cells, there was little to no basal nuclear AhR, and therefore, there were negligible effects upon the adipocyte co-cultures (Figure 3C). However, with all the cell lines, the adipocyte co-cultures significantly reduced both the *CYP1A1* and *CYP1B1* AhR target genes at 48 h. The MM.1S cells grown in transwell co-cultures with adipocytes showed elevated AhR total protein levels and significant reductions in both the *CYP1A1* and *CYP1B1* AhR target genes for both the MM.1S and MM.1R cells was observed, suggesting that a factor secreted from the adipocytes was suppressing the AhR activity (Figure 3E,F).

To determine if factors released from BMAs could prevent *CYP1A1* and *CYP1B1* transcription in the presence of an AhR agonist, the MM.1S cells co-cultured with adipocytes for 24 h were exposed to L-Kynurenine (L-Kyn; 200 μM) for an additional 24 h. The reductions in AhR total protein levels (Figure 4A) and the induction of the AhR target genes *CYP1A1* and *CYP1B1* (Figure 4B) in response to the L-Kynurenine (as observed in Figure 1) failed to occur in the presence of the adipocytes (Figure 4A). There were no significant differences in the *CYP1A1* and *CYP1B1* between the direct co-cultures with adipocytes and the direct co-cultures with adipocytes in the presence of L-Kynurenine (Figure 4B). These findings were replicated for both the MM.1R and KMS11 cells (Figure 4C,D). Interestingly, with the addition of a more potent agonist, 6-Formylindolo[3,2-b]carbazole (FICZ, 100 nM), both *CYPs* were able to be rescued to baseline levels in direct co-cultures with adipocytes (Figure 4E,F), suggesting that the BMAs blocked the AhR canonical activity in response to the exogenous agonists in a ligand-specific manner.

### 3.4. Functional Implications of AhR Repression on Multiple Myeloma Cell Proliferation

The ligand-mediated activation of AhR has been shown to act as both a promoter [39,69] and repressor [70,71,72,73] of cellular proliferation in a ligand-specific, time-frame-dependent, and ligand dosage-dependent manner. Treatment of the MM.1S cells with adipocyte-conditioned media (Figure 5A) significantly increased the cellular proliferation (Figure 5B) and increased the AhR total protein levels similar to the direct co-cultures and transwell co-cultures (Figure 1). Conversely, treatment with the AhR ligand, L-Kynurenine (200 μM), slowed the proliferation of the MM.1S cells (Figure 5B). AhR was then knocked-down using a pool of the MM.1S cells dually expressing Cas9 and either the sgRNA targeting the AhR for deletion or the empty vector control plasmid. There were observed ~50% reductions in the AhR levels in the MM.1S-Cas9 AhR knockdown (AhR KD) cells compared to the empty vector (EV) control at both the gene and protein levels (Figure 5C,D). Importantly, although the AhR protein levels were not completely diminished in the MM.1S-Cas9 AhR knockdown cells, these cells failed to induce *CYP1A1* and *CYP1B1* in response to the AhR agonists L-Kynurenine (Figure 5E) and FICZ (Appendix A). Therefore, we then measured cell proliferation via counting the viable cells over a 72 h time course (Figure 5F) to quantitatively measure the cell numbers in the MM.1S-Cas9 EV and MM.1S-Cas9 AhR KD cells. The AhR KD significantly increased the proliferation in the MM.1S cells compared to the EV controls. These findings demonstrated that AhR agonism slowed proliferation in the MM.1S cells and the AhR KD enhanced proliferation, suggesting an anti-proliferative effect of canonical AhR activity in MM cells.

## 4. Discussion

Although the classical role of the AhR is to mediate the toxicological effects of exogenous chemicals, it is also apparent that AhR has normal physiological roles in development, organ function, immunomodulation, and metabolism in response to endogenous chemicals. Whether or not endogenous chemicals can dysregulate AhR and promote pathophysiology is unknown. Herein, we described a novel mechanism in which factors from bone marrow adipocytes suppressed canonical AhR activity, leading to MM cell proliferation and highlighting a non-toxicological role of AhR in MM.

Both the dysregulation of AhR and obesity have independently been associated with a variety of different cancers; however, little is known about the role of adipocyte-secreted factors as systemic regulators of AhR in distal or neighboring cells. Huang et al. demonstrated that adipocyte-derived kynurenine activates AhR to promote adipogenesis in visceral adipose tissue [74]. Further, our laboratory recently demonstrated that BMAs secrete the endogenous AhR agonist L-Kynurenine that is taken up by epithelial cells and promotes malignant transformation [63]. Therefore, our initial hypothesis was that BMA-secreted factors would activate AhR to promote MM. In contrast to our hypothesis, we demonstrated that BMAs secrete factors that can inhibit AhR canonical activity. Similar to the AhR antagonists CH223191, alpha-Naphthoflavone, or Kyn-101, the BMAs increased AhR total protein levels (Figure 3B), decreased translocation to the nucleus (Figure 3C), and reduced the transcriptional activity of the xenobiotic phase I genes *CYP1A1* and *CYP1B1* (Figure 3D). In addition to this basal suppression of AhR activity, *CYP* induction in response to the AhR agonists was blocked in the MM cells in the co-cultures with BMAs (Figure 4) in a ligand-specific manner. BMAs represent between 50% and 70% of the bone marrow cavity [75,76,77,78] and are responsible for the secretion of a multitude of factors. BMAs have been implicated as contributing to cell growth and the progression of MM cells via the secretion and uptake of lipids and growth factors, generating a suitable environment for MM cell survival [23,37,79,80]. While the secreted mediators from BMAs that are critical for the observed phenotype are still unknown, we discovered a novel means of AhR regulation by BMAs. To our knowledge, no endogenous repressors of AhR activity have been reported. It is also likely that systemic factors released by other adipocyte depots (i.e., visceral fat) play a role in MM cell growth; however, the role of visceral adipocytes in modulating AhR activity in MM is currently unknown, and the models used herein demonstrated the direct interactions between bone marrow adipocytes, specifically, and MM cells.

Contrary to the findings herein, Bianchi-Smiraglia et al. suggested that antagonizing AhR provides a therapeutic benefit for multiple myeloma [51]. They demonstrated that Clofazamine (CLF), a drug that has been shown to have AhR antagonizing activity, induced cytotoxicity in MM cell lines, slowed MM xenograft growth, and moderately reduced disease burden in vivo by measuring serum IgG. While constitutive activation of AhR partially rescued CLF-induced death in the MM.1S cells, one limitation was that the efficacy of CLF was not tested in the AhR knockout MM.1S cells in vitro or in the xenograft model. Therefore, the AhR-independent effects of CLF could not be completely ruled out. In fact, we demonstrated that an additional antagonist, CH223191, induced cytotoxicity in both the AhR-expressing MM.1S cells and the AhR-null U266B1 cells (Appendix A). CLF, however showed better selectivity than CH223191, where the MM.1S and KMS11 cells that expressed AhR had very similar IC_50_s values (MM.1S = 1.729 µM and KMS11 = 2.168 µM) and the AhR-null U266B1 cells had an IC_50_ value of 5.109 µM (Appendix A). Collectively, these data showed that AhR antagonists are, indeed, cytotoxic to MM cells; however, the cytotoxic effects may not be fully dependent on AhR.

Pharmacological and genetic manipulations have demonstrated that AhR activity regulates MM proliferation. The AhR knockdown (KD) MM.1S cells had significantly increased proliferation compared to the EV control cells (Figure 5F), and in contrast, the AhR agonist L-Kynurenine significantly reduced proliferation (Figure 5C). Collectively, these data suggested that the inhibition of AhR activity by BMA-derived factors has the propensity to fuel MM cell growth and disease progression. This was supported by patient data demonstrating that elevated AhR gene expression levels and reduced expression levels of both *CYP1A1* and *CYP1B1* were associated with worse overall prognoses in patients (Figure 2).

## 5. Conclusions

In conclusion, these data suggested that the repression of classical AhR signaling, a phenotype driven by exposure to BMAs, is predictive of worse clinical outcomes in MM patients, supporting the evidence that MM patients with obesity have increased risk for disease progression. Future work exploring the implications of AhR signaling and its role in the pathogenesis, immune response, and etiology of multiple myeloma and its precursor conditions (monoclonal gammopathy of undetermined significance (MGUS)) and smoldering myeloma (SMM) is warranted.

## Figures and Tables

**Figure 1 cancers-15-05255-f001:**
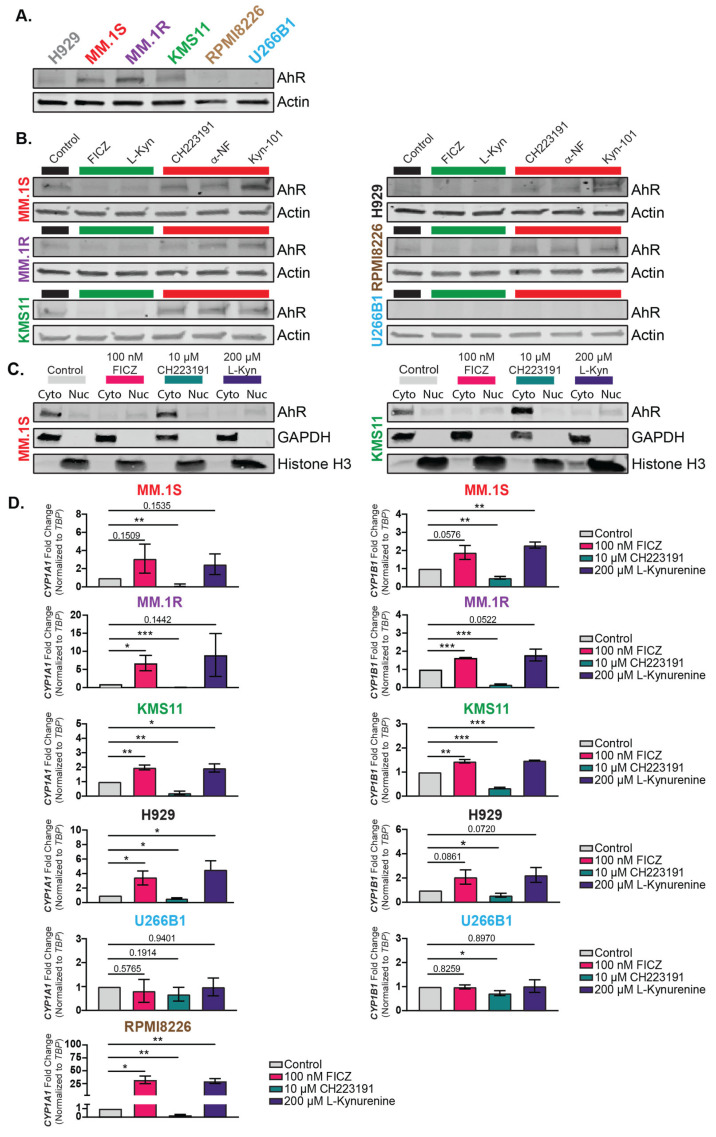
Classical AhR signaling is maintained in human MM cells exposed to AhR agonists and antagonists. (**A**) Western Blot analysis of the AhR protein levels in a panel of six multiple myeloma cell lines (H929, MM.1S, MM.1R, KMS11, RPMI8226, and U266B1). (**B**) Immunoblot analysis of AhR protein levels in response to 6-Formylindolo[3,2-b]carbazole (FICZ; 100 nM), L-Kynurenine (L-Kyn; 200 µM), CH223191 (10 µM), alpha-Napthoflavone (α-NF, 10 µM), and Kyn-101 (1 µM) for 24 h in all six human multiple myeloma cell lines. Actin was used as an internal loading control for whole-cell lysates. The agonists are depicted by the green bar and the antagonists are represented by the red bar. (**C**) Western blot analysis of AhR intracellular localization in response to the agonists FICZ and L-Kynurenine and the antagonist CH223191 in the MM.1S (**left**) and KMS11 (**right**) cells. GAPDH was used as a cytoplasmic loading control, and Histone H3 was used as a nuclear loading control. Cyto = cytoplasmic fraction and Nuc = nuclear fraction. (**D**) Taqman qPCR analysis of the target genes *CYP1A1* (**left**) and *CYP1B1* (**right**) in all MM cell lines in response to the agonists FICZ and L-Kynurenine or the antagonist CH223191. Gene expression was normalized to TBP expression as an internal loading control and shown as a fold-change relative to the vehicle-treated control. Individual *t*-tests were used to determine the significance relative to the vehicle control, and the significance values are depicted as *p* < 0.05 = *, *p* < 0.01 = **, and *p* < 0.001 = ***. The numerical *p*-values are displayed if they failed to reach significance. The uncropped blots are shown in Appendix A.

**Figure 2 cancers-15-05255-f002:**
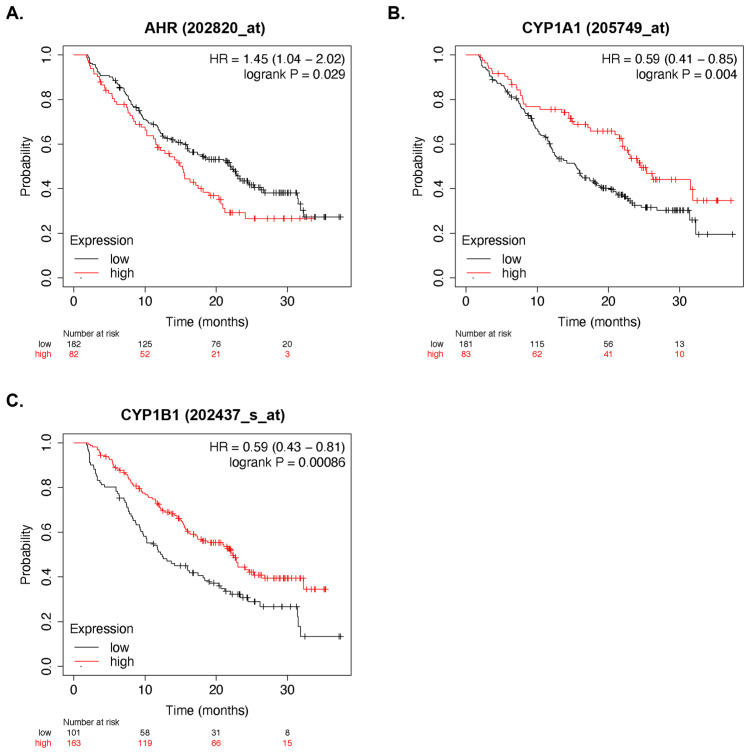
Overexpression of AhR and the decreased AhR target gene expression were associated with worse clinical outcomes. Kaplan–Meier plot of the patients expressing high *AhR* (**A**) levels, and they had worse prognoses compared to those with low expression levels. Kaplan–Meier plots of the AhR target genes *CYP1A1* (**B**) and *CYP1B1* (**C**), showing that low expression levels of the AhR target genes were associated with worse clinical outcomes. The plots were downloaded from the KM-Plotter online tool using the GSE9782 microarray database.

**Figure 3 cancers-15-05255-f003:**
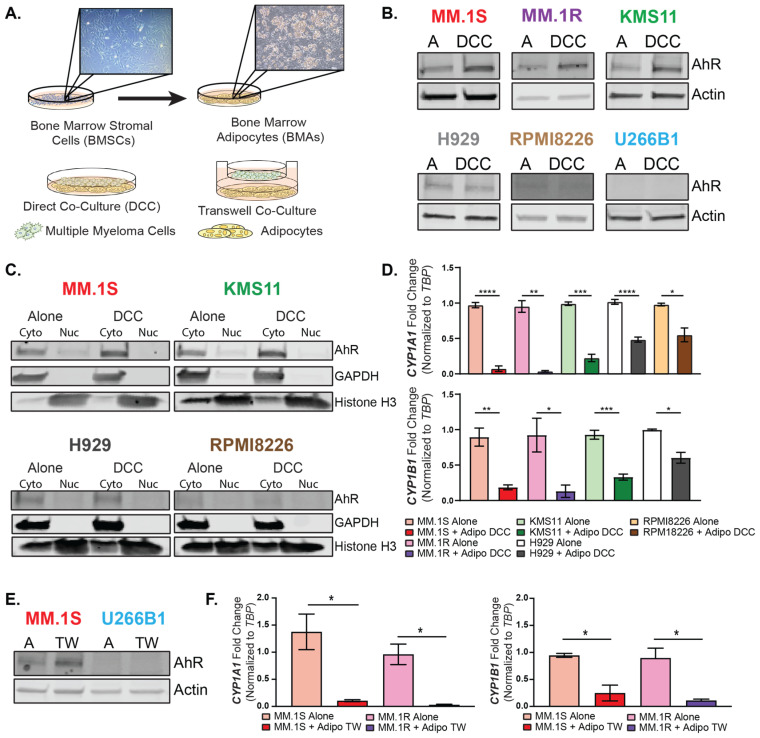
Bone marrow adipocytes repressed AhR activity in human multiple myeloma cells via secreted factors. (**A**) Schematic representation of the differentiation of bone marrow stromal cells to bone marrow adipocytes to be used in direct co-cultures (DCC) or transwell co-cultures (TW) with the MM cell lines. (**B**) Immunoblot analysis of the AhR protein levels in all six human multiple myeloma cell lines in direct co-cultures with adipocytes (DCC). (**C**) Western blot analysis of the AhR intracellular localization in response to direct co-cultures with bone marrow adipocytes in the MM.1S, KMS11, H929, and RPMI8226 cells. GAPDH was used as a cytoplasmic loading control, and Histone H3 was used as a nuclear loading control. Cyto = cytoplasmic fraction and Nuc = nuclear fraction. (**D**) Taqman qPCR of *CYP1A1* (**top**) and *CYP1B1* (**bottom**) in the panel of the MM cells alone or in direct co-cultures with adipocytes (DCC). (**E**) Western blot analysis of the AhR protein levels in the MM.1S and U266B1 cells grown alone (A) or in transwell co-cultures (TW) with bone marrow adipocytes. Actin was used as an internal loading control. (**F**) Taqman qPCR assessing the gene expression levels of *CYP1A1* (**left**) and *CYP1B1* (**right**) in the MM.1S and MM.1R cells grown in transwell co-cultures with BMAs. Individual *t*-tests were used to determine the significance levels relative to the Alone control cells, and the significance values are depicted as *p* < 0.05 = *, *p* < 0.01 = **, *p* < 0.001 = ***, and *p* < 0.0001 = ****. The uncropped blots are shown in Appendix A.

**Figure 4 cancers-15-05255-f004:**
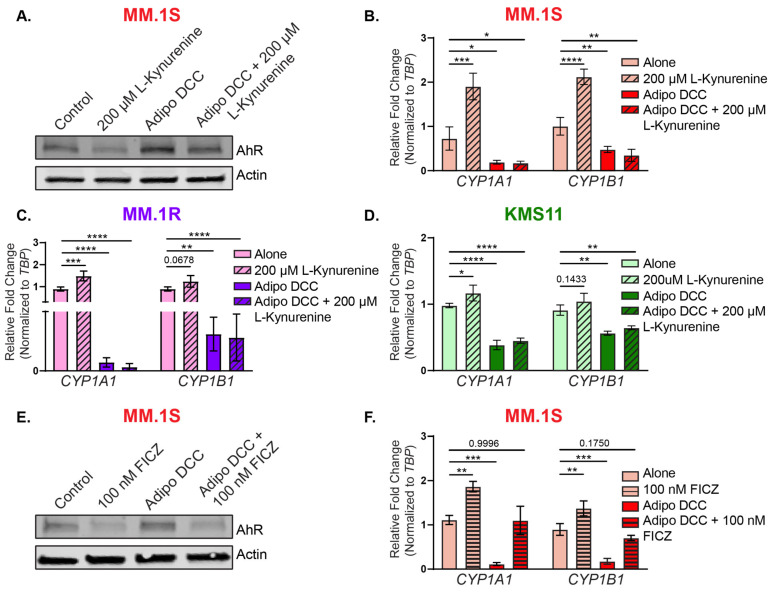
Ligand-specific competitive suppression of AhR by bone marrow adipocytes in the MM cells. (**A**) Western blot analysis of the AhR protein levels in the MM.1S cells grown alone or in direct co-cultures with bone marrow adipocytes (DCC) for 24 h and then treated with either the vehicle control or L-Kynurenine (200 µM) for an additional 24 h. Actin was used as an internal loading control. Taqman qPCR analysis of the *CYP1A1* and *CYP1B1* expression levels in the MM.1S (**B**), MM.1R (**C**), and KMS11 (**D**) cells grown alone or in direct co-cultures with bone marrow adipocytes (DCC) for 24 h and then treated with either the vehicle control or L-Kynurenine (200 µM) for an additional 24 h. Gene expression was normalized to TBP expression and shown as a fold-change relative to the untreated control cells grown in the ‘Alone’ condition. (**E**) Immunoblot analysis of the AhR protein levels in the MM.1S cells grown alone or in direct co-cultures with bone marrow adipocytes (DCC) for 24 h and then treated with either the vehicle control or FICZ (100 nM) for an additional 24 h. Actin was used as a loading control. (**F**) Taqman qPCR analysis of the *CYP1A1* and *CYP1B1* expression levels in the MM.1S cells grown alone or in direct co-cultures with bone marrow adipocytes (DCC) for 24 h and then treated with either the vehicle control or FICZ (100 nM) for an additional 24 h. Gene expression was normalized to TBP expression and shown as a fold-change relative to the untreated control cells grown in the ‘Alone’ condition. A one-way ANOVA test was used to determine the significance relative to the vehicle control grown in the ‘Alone’ condition, and the significance values are depicted as *p* < 0.05 = *, *p* < 0.01 = **, *p* < 0.001 = ***, and *p* < 0.0001 = ****. The numerical *p*-values are displayed if they failed to reach significance. The uncropped blots are shown in Appendix A.

**Figure 5 cancers-15-05255-f005:**
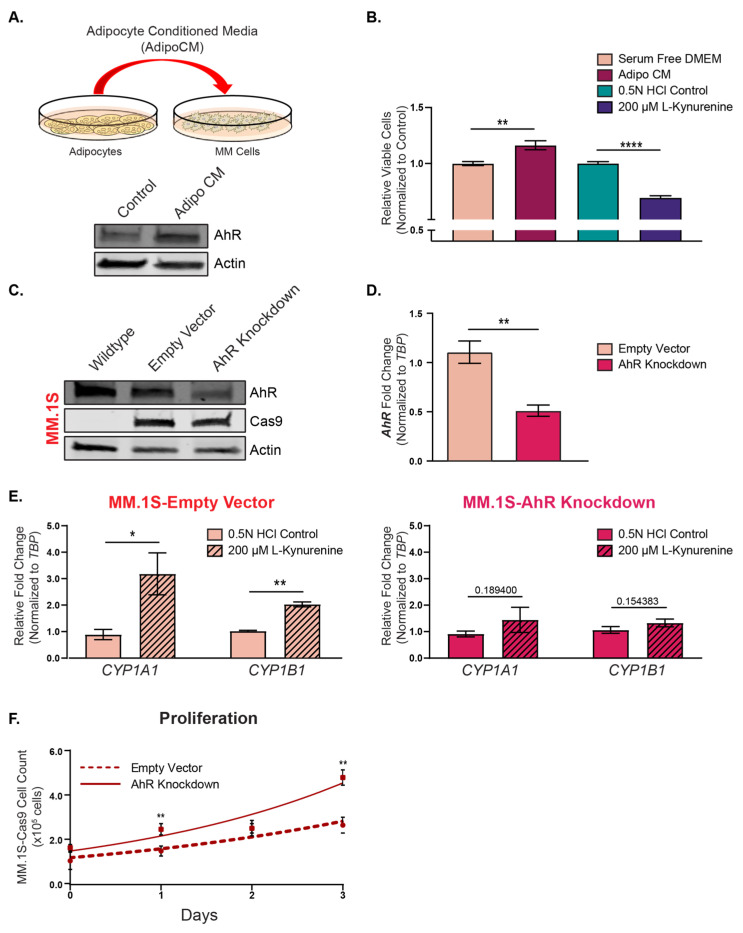
AhR activity is important for MM cell growth. (**A**) Schematic representation of the adipocyte-conditioned media treatments (**top**) and AhR immunoblot (**bottom**) in the MM.1S cells treated with serum-free DMEM (control) or the adipocyte-conditioned media (Adipo CM) for 24 h. (**B**) CellTiter-Fluor proliferation data for the MM.1S cells treated with either serum-free DMEM, the adipocyte-conditioned media, the 0.5 N HCl vehicle control, or 200 µM L-Kynurenine for 72 h. The treatment groups were plotted relative to their own controls. The data are presented as at least four technical replicates and are representative of at least two biological replicates. (**C**) Western blot analysis of the AhR protein levels in the parental MM.1S (Wildtype) cells, empty-vector-expressing MM.1S-Cas9 cells (empty vector), and AhR sgRNA-expressing MM.1S-Cas9 cells (AhR knockdown). Actin was used as a loading control. (**D**) Taqman qPCR analysis of the AhR gene expression levels in the empty vector and AhR knockdown MM.1S cells. The AhR levels were normalized to TBP expression and shown as fold-change decreases in the AhR knockdown cells relative to the empty vector cells. (**E**) Taqman qPCR analysis of the *CYP1A1* and *CYP1B1* in the empty vector (**left**) and AhR knockdown (**right**) cells in response to the L-Kynurenine (200 µM) treatment. The CYPs were normalized to TBP expression, and the results are shown as fold-changes relative to their vehicle-control-treated cells. (**F**) Trypan blue cell counts of the MM.1S empty vector cells and AhR knockdown cells over a 72 h (three days) time course. The data points represent the individual biological replicates performed in triplicate. Individual *t*-tests were used to determine the significance relative to the vehicle control cells, and the significance values are depicted as *p* < 0.05 = *, *p* < 0.01 = **, and *p* < 0.0001 = ****. The uncropped blots are shown in Appendix A.

## Data Availability

The data presented in this study are available on request from the corresponding author.

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
