# Peer review of "Non-Toxicological Role of Aryl Hydrocarbon Receptor in Obesity-Associated Multiple Myeloma Cell Growth and Survival"

_cancers, 2023, doi:10.3390/cancers15215255_

Round 1

Reviewer 1 Report

Comments and Suggestions for Authors

Congratulations to the Authors for the conceptualization of ​​the paper.

I read the entire manuscript with great interest.

The study gives us new insights into the biology of multiple myeloma. The study is highly innovative and has great clinical potential. The Authors draw attention to the need for research into the aryl hydrocarbon receptor (AhR) signaling in the context of searching new directions in multiple myeloma therapy.

However, as a Reviewer, I have a few little comments.

1. To increase the Reader's interest, it would be worth presenting in more detail the impact of obesity on the risk of progression of MM pre-cursors such as MGUS, SMM to symptomatic multiple myeloma and briefly discussing the cited works.

2. It would also be worth discussing the importance of obesity in the context of treatment outcomes.

Reviewer 2 Report

Comments and Suggestions for Authors

This is a clearly written paper, with well described experiments and appropriately presented data. I noted only a few minor technical points in the MS: FICZ should be identified as (6-Formylindolo[3,2-b]carbazole), a high affinity agonist for the AHR, and its source listed in Materials. On line 158: "…Mastermix II, no UNG…" is mysterious. In line 272 ‘data was’ should read ‘data were’. On line 398 (legend to Figure 5F, the time course is said to be 72hrs but the figure axis is labeled 1,2,3, which are presumably ‘days’. On line 424, “to [word missing?] and AHR agonists…" At the outset the authors need to provide a clear definition of what they mean by classical, toxicological and non-toxicological AHR signaling.

The results and discussion need to acknowledge that the results with AHR expression and survival of patients with MM (Figure 2A) confirm results reported by Bianchi-Smiraglia et al (authors’ reference 50), and results on expression of AHR in MM cell lines (Figure 1), including lack of AHR in U266 cells, also confirm results in the 2018 JCI paper.

There are, unfortunately, major issues with the manuscript. 1) There is a contradiction between increased AHR correlating to better overall survival, with opposite effects on expression of AHR target genes. 2) Supported by the two Supplemental Figures: The apparent 50% k/d of AHR protein expression seems to result in full loss of activity. (A fussy reviewer would also complain that the authors suggest transcriptional effects without providing any direct transcriptional activity, but this is a sloppiness of which we almost all guilty.) Anyway, there's something going on that needs explanation. 3) The assayed actions of the multifunctional AHR are limited to CYP1A1 and CYP1B1. 4) The adipocytes used for the experiments are differentiated from bone marrow stromal cells but in the absence of tumor and bone. Does this make them representative bone marrow adipocytes? 5) There is no clinical or preclinical confirmation of the results, either in an animal model or by examination of MM patient bone marrow biopsies, which should contain bone, fat and MM cells.

1  1) I would like to see a reasoned argument about the contradiction. The alternate point of view is that the contradictory effects seen in vitro from the coculture experiments suggest that the experimental approach is not physiologically relevant (a problem addressed below in 4).

22)     Would be better addressed by some additional experiments, such as readouts from a transfected reporter with XREs (AHR response elements). A problem for the field is that most of the AHR protein sits in the cytosol and very little of it actually assembles into active transcriptional complexes. So, as the authors note, protein does not equal activity, but the reader is left confused.

33)     The authors are remiss in having almost ignored reference 50, which focused on AHR regulation of the polyamine pathway in MM. The reference is just a number in a list of ones on line 82 of the Introduction that report roles for AHR in myeloma. This paper in the JCI reported extensive animal experimentation with clofazimine as an inhibitor of AHR, blocking MM growth as effectively as bortezomib. The JCI authors provide extensive evidence that the downstream effects of AHR in MM growth and progression are mediated by polyamines, including effects on the expression of ODC1 and AZIN1. Some discussion of all of this needs to be included in the MS, and the authors should include these genes in their PCR analyses.

44)   Several of the many references provided are not up to date. Fairfield et al (2023) published a detailed and complex methods article on culturing bone marrow adipocytes, which suggests to this non-specialist reviewer that this is a technologically fraught area and that the adipocytes used in the current MS may be inappropriate. They almost certainly need some detailed molecular characterization before the results obtained with MM cocultures can be convincingly evaluated.

55)     In light of the previous point, the authors really need to provide some pre/clinical correlates: either IHC of patient bone marrow slides for markers (which need to be identified and validated) or a coculture with MM and directly isolated bone marrow adipocytes or even with bone marrow/bone pieces. This latter is well beyond the aims of the MS, however. Particularly relevant is the known role of polyamines (downstream of AHR in MM) regulating the balance of osteogenesis vs adipogenesis in bone marrow (Lee et al., 2013; Tjabringa et al., 2008). Polyamines and kynurenine both regulate immune responses (Proietti et al., 2020), and MM is a disease of immune cells, but a role of the immune response in MM is not presented.

References:

Fairfield H, Condruti R, Farrell M, Di Iorio R, Gartner CA, Vary C, Reagan MR. Development and characterization of three cell culture systems to investigate the relationship between primary bone marrow adipocytes and myeloma cells. Front Oncol. 2023 Jan 11;12:912834. doi: 10.3389/fonc.2022.912834. eCollection 2022. PMID: 36713534

Lee MJ, Chen Y, Huang YP, Hsu YC, Chiang LH, Chen TY, Wang GJ. Exogenous polyamines promote osteogenic differentiation by reciprocally regulating osteogenic and adipogenic gene expression. J Cell Biochem. 2013 Dec;114(12):2718-28. doi: 10.1002/jcb.24620. PMID: 23794266

Morris EV, Edwards CM. Myeloma and marrow adiposity: Unanswered questions and future directions. Best Pract Res Clin Endocrinol Metab. 2021 Jul;35(4):101541. doi: 10.1016/j.beem.2021.101541. Epub 2021 May 1. A more recent review than authors’ reference 18, from the same group.

Proietti E, Rossini S, Grohmann U, Mondanelli G. Polyamines and Kynurenines at the Intersection of Immune Modulation. Trends Immunol. 2020 Nov;41(11):1037-1050. doi: 10.1016/j.it.2020.09.007. Epub 2020 Oct 12. PMID: 33055013

Tjabringa GS, Zandieh-Doulabi B, Helder MN, Knippenberg M, Wuisman PI, Klein-Nulend J. The polyamine spermine regulates osteogenic differentiation in adipose stem cells. J Cell Mol Med. 2008 Sep-Oct;12(5A):1710-7. doi: 10.1111/j.1582-4934.2008.00224.x. Epub 2008 Jan 11. PMID: 18194460
